# Impact of Newborn Screening and Early Dietary Management on Clinical Outcome of Patients with Long Chain 3-Hydroxyacyl-CoA Dehydrogenase Deficiency and Medium Chain Acyl-CoA Dehydrogenase Deficiency—A Retrospective Nationwide Study

**DOI:** 10.3390/nu13092925

**Published:** 2021-08-24

**Authors:** Kristina Rücklová, Eva Hrubá, Markéta Pavlíková, Petr Hanák, Martina Farolfi, Petr Chrastina, Hana Vlášková, Bohdan Kousal, Vratislav Smolka, Hana Foltenová, Tomáš Adam, David Friedecký, Pavel Ješina, Jiří Zeman, Viktor Kožich, Tomáš Honzík

**Affiliations:** 1Department of Paediatrics and Inherited Metabolic Disorders, 1st Faculty of Medicine, Charles University and General University Hospital in Prague, 128 08 Prague, Czech Republic; eva.hruba2@vfn.cz (E.H.); petr.hanak@vfn.cz (P.H.); martina.farolfi@vfn.cz (M.F.); petr.chrastina@vfn.cz (P.C.); hana.vlaskova@vfn.cz (H.V.); pavel.jesina@vfn.cz (P.J.); jiri.zeman@vfn.cz (J.Z.); viktor.kozich@vfn.cz (V.K.); 2Department of Paediatrics, 3rd Faculty of Medicine, Charles University and University Hospital Královské Vinohrady, 100 34 Prague, Czech Republic; 3Department of Probability and Mathematical Statistics, Faculty of Mathematics and Physics, Charles University, 121 16 Prague, Czech Republic; marketa@ucw.cz; 4Department of Ophthalmology, 1st Faculty of Medicine, Charles University and General University Hospital in Prague, 128 08 Prague, Czech Republic; bohdan.kousal@vfn.cz; 5Department of Paediatrics, Faculty of Medicine and Dentistry, Palacký University and University Hospital Olomouc, 779 00 Olomouc, Czech Republic; vratislav.smolka@fnol.cz (V.S.); hana.foltenova@upol.cz (H.F.); 6Institute of Molecular and Translational Medicine, Czech Advanced Technology and Research Institute (CATRIN), Palacký University Olomouc, 779 00 Olomouc, Czech Republic; tomasadam@gmail.com (T.A.); david.friedecky@upol.cz (D.F.)

**Keywords:** fatty acid oxidation disorders, neonatal screening program, clinical outcome, severity assessment

## Abstract

Long chain 3-hydroxyacyl-CoA dehydrogenase deficiency (LCHADD/MTPD) and medium chain acyl-CoA dehydrogenase deficiency (MCADD) were included in the expanded neonatal screening program (ENBS) in Czechia in 2009, allowing for the presymptomatic diagnosis and nutritional management of these patients. The aim of our study was to assess the nationwide impact of ENBS on clinical outcome. This retrospective study analysed acute events and chronic complications and their severity in pre-ENBS and post-ENBS cohorts. In total, 28 children (12 before, 16 after ENBS) were diagnosed with LCHADD/MTPD (incidence 0.8/100,000 before and 1.2/100,000 after ENBS). In the subgroup detected by ENBS, a significantly longer interval from birth to first acute encephalopathy was observed. In addition, improvement in neuropathy and cardiomyopathy (although statistically non-significant) was demonstrated in the post-ENBS subgroup. In the MCADD cohort, we included 69 patients (15 before, 54 after ENBS). The estimated incidence rose from 0.7/100,000 before to 4.3/100,000 after ENBS. We confirmed a significant decrease in the number of episodes of acute encephalopathy and lower proportion of intellectual disability after ENBS (*p* < 0.0001). The genotype–phenotype correlations suggest a new association between homozygosity for the c.1528C > G variant and more severe heart involvement in LCHADD patients.

## 1. Introduction

Long chain 3-hydroxyacyl-CoA dehydrogenase deficiency (LCHADD) and medium chain acyl-CoA dehydrogenase deficiency (MCADD) belong to the most common fatty acid β-oxidation disorders (FAOD). Upon fasting, healthy persons use β-oxidation to produce acetyl-CoA. This important intermediate can be converted to ketone bodies or serves as a source of energy in the tricarboxylic acid cycle. In addition, β-oxidation generates reduced equivalents that serve as electron donors for oxidative phosphorylation, yielding additional ATP [1]. Hence, the derangement of β-oxidation leads to an inadequate energy supply with a decreased production of ketone bodies and hypoglycaemia, as well as to an accumulation of toxic long-chain hydroxylated and medium-chain fatty acid derivatives, which induce oxidative stress and hamper multiple mitochondrial functions that contribute to tissue damage [2,3].

Both LCHADD and MCADD typically manifest with life-threatening episodes of altered consciousness, hypoketotic hypoglycaemia, liver dysfunction, and hyperammonaemia during periods of prolonged fasting or increased energy demands. Episodes of hypoglycaemia associated with brain oedema may result in permanent neurological disabilities, as described in MCADD [4]. LCHADD may, in addition, lead to rapidly worsening heart failure, attacks of rhabdomyolysis, and chronic progressive organ involvement including pigmentary retinopathy, peripheral neuropathy, and cardiomyopathy [5,6,7]. Pigmentary retinopathy starts with hypopigmentation and pigment clumping in the macula and gradually progresses to total atrophy of the posterior pole of the eye. In later stages, patients experience deteriorated night and colour vision and progressive myopia with a subsequent central vision loss [7]. Peripheral neuropathy typically begins with the loss of tendon reflexes in the lower extremities and difficulty walking on heels. Subsequently, tightness in muscles and the Achilles tendon appears to decrease the range of ankle movement. Further progression is marked by a loss of vibration sensation in lower limbs, pes cavus, calf atrophy, and gait abnormalities. Some patients may end up wheelchair bound or require surgical interventions [8].

Patients with an isolated LCHAD deficiency carry pathogenic variants in the *HADHA (Hydroxyacyl-CoA Dehydrogenase Trifunctional Multienzyme Complex Subunit Alpha)* gene. LCHAD constitutes a part of the mitochondrial trifunctional protein (MTP)—a heterotetrameric complex composed of two proteins with three enzymatic activities: Long-chain enoyl-CoA hydratase, long-chain 3-hydroxy acyl-CoA dehydrogenase, and 3-ketoacyl-CoA thiolase. Some patients suffer from an MTP deficiency (MTPD) with a deficiency of all three enzymatic functions, which is predominantly caused by pathogenic variants in the *HADHB (Hydroxyacyl-CoA Dehydrogenase Trifunctional Multienzyme Complex Subunit Beta)* gene. LCHADD and MTPD phenotypes are similar and have therefore been analysed together in this paper. MCADD is caused by pathogenic variants in the *ACADM (Acyl-CoA Dehydrogenase Medium Chain)* gene [1].

Both LCHADD/MTPD and MCADD may be identified by abnormal acylcarnitine profiles detected by tandem mass spectrometry in neonatal screening (NBS) programs, allowing for a presymptomatic diagnosis and the management of these patients. The mainstay of treatment consists of dietary interventions such as the avoidance of fasting and administration of carbohydrates during increased metabolic stress. LCHADD/MTPD patients also require a fat-restricted diet and medium chain triglyceride (MCT) oil supplementation [9].

Previous studies have shown that the implementation of NBS and early dietary management reduces mortality [10,11,12] as well as the risk of developmental disability [13] in MCADD patients, while the evidence for the benefits of expanded NBS (ENBS) for LCHADD/MTPD patients is rather limited [14]. In fact, chronic complications of LCHADD/MTPD such as retinopathy and neuropathy have been reported to remain irreversible and progressive, despite early dietary interventions [6].

The aim of our study was to analyse the impact of ENBS and early initiated nutritional management on clinical outcome in patients with LCHADD/MTPD and MCADD compared to the pre-ENBS period in a nationwide study covering more than three decades.

## 2. Patients and Methods

### 2.1. Patients

Almost all patients with LCHADD/MTPD (*n* = 28) and MCADD (*n* = 69) who were born between 1984 and 2020 in Czechia, and whose diagnosis was genetically confirmed, were included in the study. Three patients with clinical and biochemical findings consistent with LCHADD/MTPD were excluded because they did not sign their informed consent with genetic testing (*n* = 2 died, *n* = 1 before ENBS, and *n* = 1 after introduction of ENBS). Seven patients with MCADD, who were diagnosed presymptomatically beyond the neonatal period due to family screening in affected families in the prescreening era were excluded from the clinical outcome study but were included in the calculation of incidence.

The diagnosis was based on blood acylcarnitine and urinary organic acid profile and confirmed by molecular genetic testing. Until 2011, molecular genetic analysis was performed by PCR/RFLP methods to detect the most prevalent mutations in the *HADHA* (c.1528C > G) and the *ACADM* (c.985A > G) genes. Since 2011, the *ACADM* and *HADHA* genes were analysed by Sanger sequencing of PCR products of all coding exons. Samples that had been analysed before 2011 and in which no prevalent mutation had been detected were reanalysed by Sanger sequencing after 2011. The *HADHB* gene was analysed by massive parallel sequencing within a panel of metabolic disorders. The observed genetic variants were described according to the HGVS nomenclature using the *ACADM* or *HADHA/HADHB* reference DNA/RNA sequences GenBank: *ACADM*-NC_000001.11 (75724709..75763679) 16-MAY-2021, NM_000016.6, 20-APR-2021, *HADHA*-NC_000002.12 (26190635..26244632 complement), 16-MAY-2021, NM_000182.5, 26-JUN-2021, and *HADHB*- NC_000002.12 (26244917..26290465) 16-MAY-2021, NM_000183.3, 06-MAY-2021. Pathogenicity of novel variants was evaluated by an in-silico analysis using PolyPhen-2, Mutation Taster [15,16] and VarSome [17].

As soon as the diagnosis was suspected, based on acylcarnitine profiles, the parents of the patients were advised to adhere to current nutritional recommendations [9]. In LCHADD/MTPD, the principle of management is to avoid fasting and limit the intake of long-chain fatty acids (LCFA) while supplementing MCT, which provides an alternative energy source downstream of the enzymatic block and further decreases LCFA oxidation. The maximum recommended fasting period is 3 h for the age group 0–3 years, 4–5 h for preschool and school children, and 6 h for adolescents and adults. Patients have been instructed to preferentially eat foods containing complex carbohydrates (corn starch and other starchy products) and avoid diets high in natural fat. MCT should cover 20% and LCFA only 10% (less than 1.0 g/kg/day) of total energy intake with an adequate amount of essential fatty acids (3–4%). Cooperation with most of our patients/parents was excellent and their compliance with the diet was good. The dietary recommendation was the same in both pre-ENBS and post-ENBS groups. In recent years, an odd-carbon (C7) chain triglyceride (triheptanoin), which offers a promising alternative to standard even-carbon (C8) MCT oil has been introduced, which has proven more efficient in reducing acute events such as hypoglycaemia and rhabdomyolysis and in improving cardiac function [18,19,20,21] than the standard MCT oil. As triheptanoin is not available in our country, only one of our patients was switched to this product in 2019 in a compassionate use program.

In MCADD, the main goal is to prevent hypoglycaemia. The maximum recommended fasting period varies with age, i.e., 3 h for newborns and infants, 4 h for toddlers, 5 h for preschool children, 6–7 h for school children, and 8 h for adolescents and adults. Patients with MCADD have been also recommended to consume frequent meals with high complex carbohydrate content, adhere to a low-fat diet (30% of total energy), and avoid meals containing MCT oil (e.g., coconut oil).

All patients were regularly followed in two tertiary care hospitals—General University Hospital in Prague and the University Hospital Olomouc. Their follow-up included the regular clinical, metabolic, and psychological assessment of the MCADD patients and additional regular cardiologic, ophthalmologic, and neurologic examination of the LCHADD/MTPD patients.

### 2.2. Methods

For comparison of clinical outcome LCHADD/MTPD and MCADD, patients were subdivided into those clinically ascertained in the pre-ENBS period and those diagnosed presymptomatically in the neonatal period by ENBS. The patients’ records were evaluated, and relevant data were extracted by three investigators. Another independent investigator checked the accuracy of collected data. Both disease groups were analysed separately with a focus on acute events and chronic complications. In addition, in LCHADD/MTPD patients’ anthropometric data, including height, weight, and BMI, were collected and compared with reference ranges for our population [22].

#### 2.2.1. LCHADD/MTPD Acute Events

Death, all episodes of altered consciousness/acute encephalopathy, attacks of rhabdomyolysis, and acute heart failure requiring inotropes were recorded for each patient together with the date of the respective event. Rhabdomyolysis was defined as a combination of acute muscle weakness and/or pain together with a creatine kinase (CK) elevation of at least 5 times the upper limit of the reference range of 11 µkat/L, which corresponds to 660 IU/L [23].

#### 2.2.2. LCHADD/MTPD Chronic Complications

Presence and progression of retinopathy, peripheral neuropathy, and intellectual impairment were assessed during the follow-up. Staging of retinopathy was based on eye fundoscopy and vision assessment in agreement with previously published criteria [7]. Cognitive functioning was assessed using standardized psychological tests such as the Gesell developmental scale, Stanford–Binet intelligence scale, and Wechsler intelligence scale, as appropriate. Cardiomyopathy was diagnosed based on echocardiographic criteria [24]. Hypertrophic cardiomyopathy was defined by maximum left ventricular wall thickness of >2 Z scores [25]. Mixed phenotype was diagnosed in cases of phenotypic overlap—for instance, in cases where hypertrophy was associated with decreased systolic function, etc.

Severity score as defined in Appendix A was assigned to acute and chronic complications. Scoring of retinopathy and rhabdomyolysis was based on published reports [7,26,27,28] and severity of intellectual impairment corresponded to the DSM-IV classification.

#### 2.2.3. MCADD

Death, episodes of altered consciousness/acute encephalopathy, and intellectual impairment and their severity were assessed as described above for LCHADD/MTPD patients.

#### 2.2.4. Statistical Analysis

Incidence rates were calculated based on the number of newborns born in 1998–2009 (1,206,358 in total, before ENBS) and in 2010–2020 (1.225,245 in total, after ENBS) as provided by the Czech Statistical Office (www.czso.cz (accessed on 14 June 2021)), and the number of patients with a confirmed diagnosis (10 and 15 with LCHADD/MTPD and 9 and 53 with MCADD) who were born during these two periods, respectively.

Continuous data were summarized as means with standard deviation (SD) and/or as medians with range and interquartile range (IQR). Categorical data are presented using absolute and relative frequencies.

For various acute and chronic complications, including death, the probability of symptom-free period from birth and its association with ENBS was explored through the Kaplan–Meier estimator and tested using log-rank test. In patients diagnosed in the pre-ENBS period, there were no events beyond 16.5 years of age, and curves were truncated at 17 years of follow-up. Two patients (one with MCADD and one with LCHADD) were detected by a pilot screening study conducted between 2001 and 2009 and were included in the post-ENBS cohort. Therefore, the follow-up of the post-ENBS subgroup extended beyond 12 years.

Recurring acute events were analysed using the incidence rate (IR), expressed as the number of events per one patient-year. The association of IR with ENBS was tested using the incidence rate ratio (IRR), with IR for the ENBS group in the numerator and for the pre-ENBS group in the denominator. The 95% confidence limits for the IRR and the corresponding *p*-value were determined using the appropriate χ^2^-test. Similarly, the severity of acute events was determined by calculating the sum of severity score for each subgroup per one patient-year of follow-up.

We have also considered possible differences between patients in respect to the progression of chronic complications. Our motivation was to discern differences between a patient that developed a chronic complication of certain severity early in life and a patient that developed the same degree of the respective complication later in life. We have therefore computed areas under curve (AUC) from a time vs. severity score plot for each patient and divided it by the follow-up period (FUP). Examples for various patients with retinopathy are given in Appendix A. The resulting ratio was then compared between the pre-ENBS and post-ENBS subgroups using two-sample t-tests.

In addition, we compared the incidence rate of acute events and the severity of chronic complications in patients homozygous for the most prevalent variants in the *HADHA/HADHB* and *ACADM* genes vs. patients with other genotypes to reveal possible genotype–phenotype correlations. Finally, Fisher’s exact test was used to evaluate the relationships between acute events and the development of chronic complications.

Statistical language and environment R, version 4.1, was used throughout the analysis. Libraries *survival* and *epiR* were used for statistical testing. The level of statistical significance was set to 0.05.

## 3. Results

### 3.1. LCHADD/MTPD and MCADD Demographic Data

Basic characteristics of LCHADD/MTPD and MCADD cohorts are provided in Table 1.

### 3.2. LCHADD/MTPD and MCADD Genotypes

In LCHADD/MTPD patients, 10 different pathogenic or likely pathogenic variants were detected in the *HADHA* as well as in the *HADHB* genes. Forty alleles (19/24 identified before and 21/32 after ENBS) contained the most prevalent c.1528G > C variant. Three variants in the *HADHA* gene were novel [c. (?_-1)_(975+1_976-1)del (deletion of exons 1-10), c.58delC, c.67+2986_315-848del, c.799+5_799+17del] and one in the *HADHB* gene [c.(1389+1_1390-1)_(*1_?)del (deletion of about 6700 bp-from exon 16 to 3’UTR)].

In MCADD patients, 17 different pathogenic variants were detected in the *ACADM* gene. One-hundred and six alleles (28/30 identified before ENBS and 78/108 identified by ENBS) carried the most common c.985A > G variant. Three of the detected variants were novel [c.31-1323_118+923del, c.387G > A and c.(945+1_946-1)_(*1_?)del (deletion of exons 11–12)]. The distribution of genetic variants is depicted in Figure 1A,B.

The predicted effect of all variants at the mRNA and/or protein level is shown in Appendix A.

### 3.3. LCHADD/MTPD Anthropometric Parameters

Height and BMI during follow-up of patients with LCHADD/MTPD were plotted and compared to normal values. The height of eight patients from both subgroups was below −2 Z scores, which was related to prematurity in six of them, as they exhibited a sufficient catch-up growth within the first year of life. Only two children remained below −2 Z with respect to height during long-term follow-up. BMI exceeded +2 Z in five children (with maximum of +5.6 in one girl) during long-term follow-up (Figure 2A,B).

### 3.4. LCHADD/MTPD and MCADD Clinical Outcome

#### 3.4.1. LCHADD/MTPD

Mortality as well as acute and chronic complications were analysed in order to compare the clinical outcome of patients diagnosed based on symptoms before ENBS with those detected by ENBS. Results are shown in Table 2.

Mortality after ENBS did not decrease significantly. Among patients diagnosed before ENBS, five (three boys, two girls) children died, in four of them the diagnosis of LCHADD/MTPD was established postmortem. The three boys died suddenly and unexpectedly at home and the two girls died in hospital due to heart failure during acute metabolic decompensation preceded by vomiting. The median age at death was 7 months (range 1.25–72). In the subgroup detected by ENBS two children (one girl, one boy) died from an acute heart failure at the age of 20 months and from multi-organ failure during gastroenteritis at the age of 3.5 years, respectively (median 31 months). Hence, death occurred later in the post-ENBS subgroup.

Number and severity of acute events did not differ significantly between the two subgroups with the exception of episodes of severe and critical rhabdomyolysis that were significantly more frequent and more severe in patients diagnosed by ENBS. Interestingly, one of our patients diagnosed in the era before ENBS experienced 83 attacks of rhabdomyolysis during 15 years of follow-up and also suffered from the most severe neuropathy.

Severity of chronic complications was assessed using the area under curve obtained from time vs. severity score plots divided by patient-years of follow up (Appendix A) for pre-ENBS and post-ENBS cohorts. No statistically significant improvement was found after ENBS, although certain positive trends may be present for neuropathy and cardiomyopathy (*p* = 0.07 and *p* = 0.09, respectively). Overall severity of retinopathy and intellectual disability did not differ between the pre-ENBS and post-ENBS subgroup.

Normal eye fundoscopy was described in only four patients (two before and two after ENBS), their age at last visit was 14, 15, and 3 years and 7 months. Both of the oldest patients carried the c.703C > T variant on at least one allele. Retinopathy was mild in all but two of the remaining patients, who exhibited a moderate and severe retinal involvement with deteriorated vision. They were both diagnosed by ENBS and their retinopathy progressed to the moderate stage at the age of six and eight years, respectively, and deteriorated to a severe grade by 15 years of age in one of them.

Peripheral neuropathy was absent in six patients diagnosed after ENBS, their median age at the last visit was 3.15 years (range 0.67–9.75). Severe neuropathy necessitating orthopaedic surgery and wheelchair use was present in only one boy who was diagnosed clinically at the age of 2.5 years and his neuropathy deteriorated to a severe stage at the age of nine years. He was a homozygote for c.703C > T variant in the *HADHA* gene. His ophthalmologic and cardiologic findings were completely normal throughout 14 years of follow-up. Progression to moderate neuropathy was observed in two patients detected before and four after ENBS at a median age of 4.75 years (range 2.5–7) and 8.13 years (range 5.75–10), respectively. One of these patients required surgical prolongation of Achilles tendons at the age of nine years.

Cardiomyopathy subtypes observed in the pre-ENBS subgroup included hypertrophic, dilated, non-compaction, and hypertrophic evolving into restrictive pathophysiology in 5, 1, 1, and 1 child, respectively. In two out of five patients with myocardial hypertrophy, the hypertrophy regressed completely, while all other patients with cardiomyopathy died except for one subject with a restrictive subtype, who was switched to triheptanoin oil in 2019. Since the initiation of this alternative treatment, his heart function has remained stable. In the subgroup of patients diagnosed by ENBS, four children manifested transient hypertrophic cardiomyopathy. One child suffered from dilated cardiomyopathy and eventually died during an acute decompensation. The last patient’s phenotype evolved from hypertrophic to mixed hypertrophic/dilated and later improved to an almost normal cardiac morphology with a mild residual mitral valve insufficiency.

Intellectual disability was observed in five patients, two of which had a severe neurologic impairment related rather to severe perinatal asphyxia. The remaining three patients were only mildly affected. All of them had a history of at least one episode of severe hypoglycaemia with loss of consciousness.

Survival probability and probability of absence of symptoms is shown in Figure 3A–D for acute events and in Figure 4A–D for chronic complications.

The only statistically significant improvement after ENBS was the longer symptom-free interval from birth to the first episode of acute encephalopathy.

#### 3.4.2. MCADD

The mortality, number, and severity of episodes of altered consciousness/acute encephalopathy and cognitive impairment were analysed to compare the clinical outcome of patients diagnosed in the pre-ENBS vs. post-ENBS era. Results are shown in Table 3.

Our study failed to demonstrate a significant decrease in mortality as the overall mortality in both subgroups was relatively low. One girl died in the post-ENBS era at the age of three-days-old before the results of ENBS were available. Two patients (one boy, one girl) died in the pre-ENBS era during acute gastroenteritis at the age of 14 months and suddenly at 6.5 years, respectively. A sibling of one of them had died previously after 3 days of life, presumably due to unrecognized MCADD. On the other hand, our study proved a significant drop in the number and severity of episodes of acute encephalopathy that were almost invariably associated with hypoglycaemia. It also showed a significant decrease in the incidence and severity of cognitive impairment. Intellectual disability was defined by DSM-IV. Two children from the whole cohort (both detected before ENBS) suffered from moderate intellectual disability and the rest were only borderline or mildly affected. Nobody was severely handicapped.

Survival probability and freedom from acute encephalopathy and intellectual disability are shown in Figure 5A–C.

### 3.5. Relationship between Acute Events and Chronic Complications

In LCHADD/MTPD, we explored the influence of rhabdomyolysis and acute heart failure on the development of cardiomyopathy, and we found no significant relationship, with *p* = 1 and *p* = 0.077, respectively. On the other hand, the number and severity of acute encephalopathy episodes were significantly associated with the presence of cardiomyopathy, with *p* = 0.007 and 0.009, respectively. Peripheral neuropathy was found to be related to the number and severity of all acute events, with *p* = 0.003 and 0.001, respectively.

In MCADD, we confirmed a statistically significant relationship between intellectual disability and the number and severity of episodes of acute encephalopathy, with *p* = 0.0003 and *p* < 0.0001, respectively.

### 3.6. Genotype-Phenotype Correlation

In the LCHADD/MTPD cohort, homozygotes for c.1528G > C variant were compared to patients with other genotypes. Homozygotes experienced statistically more episodes of heart failure (*p* = 0.005) and were more affected by cardiomyopathy (*p* = 0.01). Their episodes of altered consciousness were significantly more severe (*p* = 0.01). On the other hand, they suffered from significantly less attacks of rhabdomyolysis (*p* = 0.068) that were less severe (*p* = 0.04). No other significant genotype–phenotype correlations were identified. Acylcarnitine profiles did not differ significantly with respect to genotype.

Similarly, in the MCADD cohort, homozygotes for c.985A > G variant were compared to the rest of the patients. The number (*p* = 0.007) and severity of episodes with altered consciousness (*p* = 0.001) was significantly higher in homozygotes, while the severity of intellectual impairment did not differ significantly (*p* = 0.10). Regarding acylcarnitine profiles, only the C8/C10 ratio was significantly higher in homozygotes diagnosed by ENBS (*p* = 0.007) as opposed to patients with other genotypes in the ENBS era.

## 4. Discussion

Following a pilot study conducted in 2001–2009, the Czech national NBS program was expanded to include LCHADD/MTPD and MCADD on October 1st, 2009. Since then, all children have been detected presymptomatically and their parents have been advised to follow current dietary recommendations soon after the birth of the child. The families have had regular consultations with specialized metabolic nutritionists, not only to prevent symptoms but also to achieve optimum growth without excessive weight gain due to frequent feedings. For this reason, we investigated anthropometric data in our LCHADD/MTPD cohort. We observed good growth and weight control in our patients. Only two (2/28) grew below normal range and five (5/28) had a Z score for BMI > +2, which is lower than in a Swedish study that reported overweight or obesity in the majority (8/10) of patients [29].

In agreement with previous studies [12,30] we confirmed an increased incidence and genetic heterogeneity of both LCHADD/MTPD and MCADD after the introduction of ENBS. The estimated incidence of LCHADD/MTPD and MCADD increased from 0.8/100,000 and 0.7/100,000 before to 1.2/100,000 and 4.3/100,000 after ENBS, respectively. These findings indicate that some patients may have been missed and died undiagnosed in the pre-ENBS era. Another possible reason might be that ENBS may detect milder forms of the disease, possibly resulting in unnecessary dietary restrictions in patients who may never become symptomatic [30]. For instance, c.199T > C variant in the *ACADM* gene has been overrepresented in post-ENBS cohorts and was considered mild with residual enzyme activity in compound heterozygotes within the range of healthy individuals [31]. In agreement with these findings, we had four compound heterozygotes with this variant (all in the post-ENBS subgroup) and they were asymptomatic. On the other hand, Gramer et al. [32] described two compound heterozygous patients carrying this variant who suffered from neonatal hypoglycaemia with altered consciousness. Hence, dietary restrictions in these patients have been alleviated, but not completely abandoned.

Focusing on clinical outcomes in patients diagnosed before vs. after ENBS, our study is relatively unique in its complexity because it includes multiple outcome parameters such as mortality, acute events, as well as chronic complications. In addition, it also provides a comparison of the severity of the respective complications. The calculation of area under curves obtained from a time versus severity score plots, which were used to assess and compare the severity of chronic complications in LCHADD/MTPD patients, may be used as a practical tool for similar studies. Moreover, our study covers an exceptionally long follow-up period of more than 200 patient-years and more than 500 patient-years for LCHADD/MTPD and MCADD, respectively.

In the LCHADD/MTPD cohort, we failed to demonstrate any significant improvement in mortality and overall clinical outcome after ENBS. The only findings suggesting a benefit of ENBS were significantly longer interval from birth to the first acute encephalopathy and a non-significantly better outcome with respect to neuropathy and cardiomyopathy. On the other hand, attacks of severe and critical rhabdomyolysis were more common and more severe in patients diagnosed after ENBS.

Previous studies have demonstrated lower mortality in early treated subjects ranging from 0 to 20% as compared to 23–50% in late treated subjects [8,9,33,34]. However, only in one of these studies was the decrease statistically significant [33]. Some studies reported a significant decrease in episodes of hypoglycaemia during follow-up from 88 to 40% after ENBS [9], whereas other studies showed non-significant differences [35] or were limited to patients diagnosed clinically [36]. In contrast to our findings, previous studies reported that more patients (100%) diagnosed on clinical grounds experienced rhabdomyolysis than those detected by ENBS (82–86%) [37,38] though the difference was not statistically significant. Our paradoxical findings could be possibly explained by a lack of complete medical records, including CK levels, in the era before ENBS.

The analysis of chronic complications in LCHADD/MTPD did not demonstrate any significant benefit of ENBS. However, an analysis of retinopathy and neuropathy may be skewed by missing ophthalmologic and neurologic examinations in the pre-ENBS era, as opposed to a more systematic ophthalmologic and neurologic follow-up in the post-ENBS era, reflecting increasing knowledge about these complications. Previous studies assessing the impact of ENBS on chronic complications are rather scarce. Some have reported on retinopathy in screened vs. unscreened patients, but the difference was not statistically significant [8,35,39,40]. Only one study found a statistically significant decrease in retinopathy from 100% (5/5) to 33.3% (3/9) after ENBS [39]. In contrast, our study revealed that retinopathy was relatively more common in patients detected in the ENBS era, with 84.6% (11/13) vs. 75% (6/8) in the pre-ENBS era. Neuropathy in screened vs. unscreened patients was analysed only in one study, which reported 0% (0/1) screened and 33.3% (3/9) unscreened patients affected [8], compared to 62.5% (10/16) and 100 (7/7) in our cohort, respectively. The progression of neuropathy during follow-up had previously been studied by EMG in screened patients and revealed first signs of neuropathy between 6 and 12 years of age [41], which is later than in our cohort (median 2.91 years). This might be explained by a low threshold for detecting neuropathy in our cohort, as we considered decreased tendon reflexes in lower extremities as signs of neuropathy. Unfortunately, EMG was not available in most of our patients.

Cardiomyopathy represents another chronic, potentially life-threatening complication of LCHADD/MTPD. We observed a non-significant reduction in prevalence of cardiomyopathy after ENBS. In addition, most (5/6) patients with cardiomyopathy in the ENBS-detected subgroup have shown a complete regression of cardiomyopathy during follow-up, and only one patient with cardiomyopathy died during an acute metabolic crisis. In contrast, five out of eight patients with cardiomyopathy in the pre-ENBS era died, one has persisting severe cardiomyopathy with restrictive pathophysiology, and, in two patients, echocardiography findings regressed to normal during follow-up. Our findings contrast with a previous study which demonstrated a statistically significant difference in favour of ENBS, with three out of five vs. zero out of seven affected patients, respectively [21]. Improvement reported by other studies was not statistically significant with 0–40% in screened vs. 25–100% in the unscreened subgroup during a follow-up of 2 to 20.5 years [5,8,9,21].

Due to lack of evidence for any unambiguous benefit of ENBS for LCHADD/MTPD patients, some countries (e.g., Belgium, Luxembourg, Switzerland, and the UK) have not included LCHADD/MPTD in their neonatal screening programs [42]. Future routine use of the more efficient MCT oil, triheptanoin, may result in a better outcome for LCHADD/MTPD patients than reported in our study [43].

As opposed to LCHADD/MTPD, a beneficial effect of ENBS on mortality and morbidity in patients with MCADD has been well established [44]. Mortality and the prevalence of episodes of acute metabolic decompensation have been reported to decrease significantly from 14–21% to 3–4% and from 66–95% to 0–13%, respectively [4,10]. In agreement with most studies, we confirmed a significant decrease in the number and severity of acute metabolic derangements in our cohort. On the other hand, we failed to demonstrate a significant reduction in mortality, probably due to undiagnosed deceased patients in the pre-ENBS era. In agreement with previous studies, we showed a significant decrease in the prevalence and severity of cognitive impairment from 67% before to 11% after ENBS, as compared to 10–40% vs. less than 3% reported previously [4,45,46]. Higher numbers in our study may be related to the fact that, in contrast to previous studies, we included borderline and mild cognitive impairment as well. If we restricted the analysis to moderate and severe intellectual disability, our results would be similar with 13% (2/15) before and 0% (0/54) after ENBS.

Apart from clinical outcome in patients detected in the pre- and post-ENBS era, we also analysed genotype–phenotype correlations and the relationship between acute events and chronic complications. We confirmed a significant association between the number of episodes of altered consciousness and cognitive impairment in MCADD, as reported previously [4]. In LCHADD/MTPD, we failed to prove the expected relationship between the number of rhabdomyolysis or episodes of acute heart failure and the development of cardiomyopathy. On the other hand, we found an association between episodes of altered consciousness and cardiomyopathy, as well as between all acute events and neuropathy. We suggest that this might be explained by a more severe phenotype in general, rather than by genuine causality.

Considering genotype–phenotype correlations, we demonstrated that LCHADD/MTPD homozygotes for the c.1528G > C variant manifested more episodes of acute heart failure and were more often affected by cardiomyopathy. At the same time, they suffered from significantly less attacks of rhabdomyolysis that were less severe. To the best of our knowledge, no such observations have been published before. Nevertheless, our results may be distorted by the fact that rhabdomyolysis could have escaped detection in the era before ENBS, and that much greater proportion of patients diagnosed with LCHADD/MTPD before ENBS were homozygotes for c.1528G > C variant in comparison to the subgroup after ENBS. On the other hand, we did not confirm previous observations that homozygotes are more frequently affected by retinopathy [47]. In agreement with other reports [48,49], homozygotes for the c.985A > G variant in our MCADD cohort manifested a more severe phenotype with more frequent and serious episodes of altered consciousness than the patients with other genotypes. All of the mentioned reports suggested an association between homozygosity for c.985A > G and higher C8 values and C8/C2 and C8/C10 ratios. Our study, on the contrary, showed no significant relationship between acylcarnitine profiles and genotype, neither in MCADD nor in LCHADD/MTPD, with the exception of a higher C8/C10 ratio in the MCADD homozygotes diagnosed by ENBS (*p* = 0.007).

### Limitations

A lack of complete medical records in patients diagnosed in the pre-ENBS era and a small cohort size, especially of LCHADD/MTPD patients, are the major limitations of our study.

## 5. Conclusions

Our study confirmed a clear benefit of ENBS for MCADD patients, as it reduced the number and severity of episodes of acute encephalopathy and decreased the prevalence and severity of intellectual disability. On the other hand, we failed to demonstrate an improved outcome in LCHADD/MTPD patients, although a certain positive trend in neuropathy and cardiomyopathy may be present together with a longer symptom-free period from birth to the first acute encephalopathy. In addition, new insights into genotype–phenotype correlations, such as a higher susceptibility to heart involvement in homozygotes for the c.1528G > C variant in the *HADHA* gene, are suggested here. The present study is unique in its complexity, covering multiple outcome parameters together with their severity, and some of the statistical methods for the evaluation of chronic complications may be useful for similar studies in the future.

## Figures and Tables

**Figure 1 nutrients-13-02925-f001:**
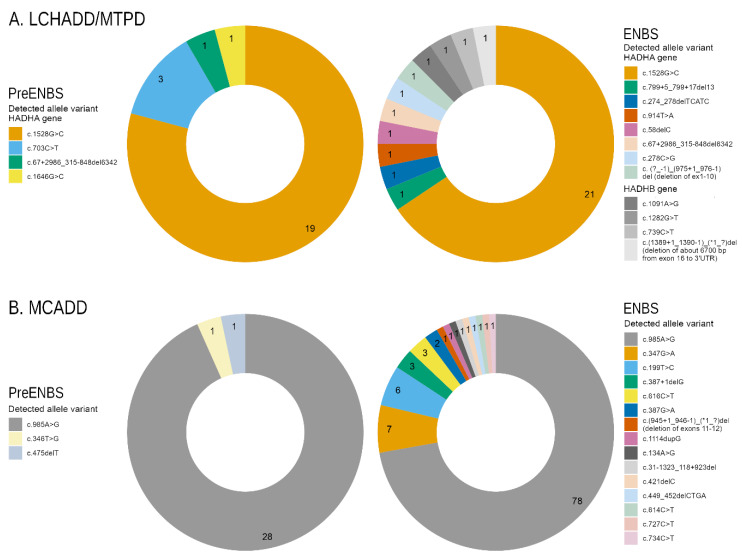
Distribution of genetic variants: (**A**) Graphs show proportion of alleles in *HADHA* and *HADHB* genes in patients diagnosed with LCHADD/MTPD before (*n* = 12) and after ENBS (*n* = 16). (**B**) Graphs show proportion of *ACADM* alleles in MCADD patients before (*n* = 15) and after ENBS (*n* = 54). Details on genotypes are shown in Appendix A. LCHADD/MTPD: Long chain 3-hydroxyacyl-CoA dehydrogenase deficiency/ mitochondrial trifunctional protein deficiency; MCADD: medium chain acyl-CoA dehydrogenase deficiency; pre-ENBS: cohort diagnosed before introduction of expanded neonatal screening; ENBS: cohort diagnosed after introduction of neonatal screening; HADHA: Hydroxyacyl-CoA Dehydrogenase Trifunctional Multienzyme Complex Subunit Alpha; HADHB: Hydroxyacyl-CoA Dehydrogenase Trifunctional Multienzyme Complex Subunit Beta; ACADM: Acyl-CoA Dehydrogenase Medium Chain.

**Figure 2 nutrients-13-02925-f002:**
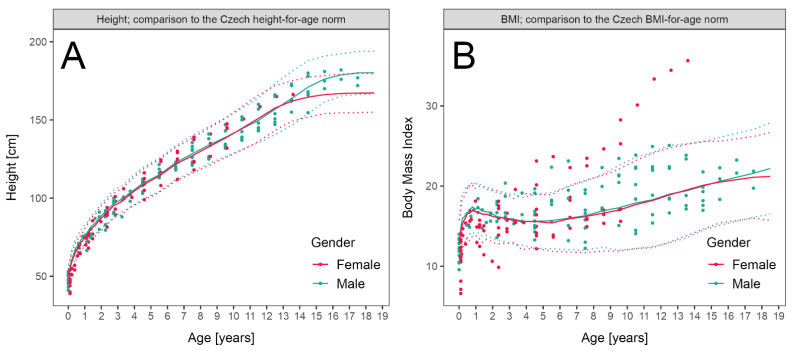
Anthropometric parameters in patients with LCHADD/MTPD: (**A**) Height of patients diagnosed before ENBS (data available in 10 patients, 111 measurements, median 11 measurements per patient) and by ENBS (*n* = 16, 95 measurements, median 6.5 measurements per patient) during follow-up compared to reference range. (**B**) Body Mass Index of patients diagnosed before ENBS (*n* = 12, 111 measurements, median 11 measurements per patient) and by ENBS (*n* = 16, 95 measurements, median 6.5 measurements per patient) during follow-up. Patient data are shown as individual points, median and 2–98th centile of sex-adjusted reference ranges for the Czech population are shown as solid and dotted lines, respectively.

**Figure 3 nutrients-13-02925-f003:**
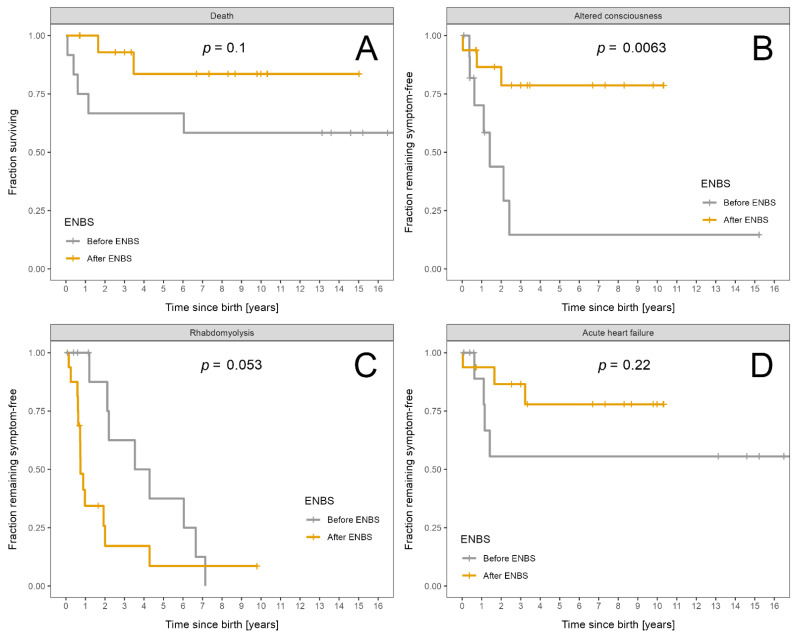
Fraction of LCHADD/MTPD patients remaining alive and symptom free: Patients diagnosed before ENBS (*n* = 12) are shown as grey and patients diagnosed after ENBS (*n* = 16) as orange curves, respectively. (**A**) Survival probability. (**B**) Absence of any episode of altered consciousness. (**C**) Absence of rhabdomyolysis. (**D**) Absence of acute heart failure. Kaplan–Meier estimator was used; *p* designates significance of difference between the cohorts.

**Figure 4 nutrients-13-02925-f004:**
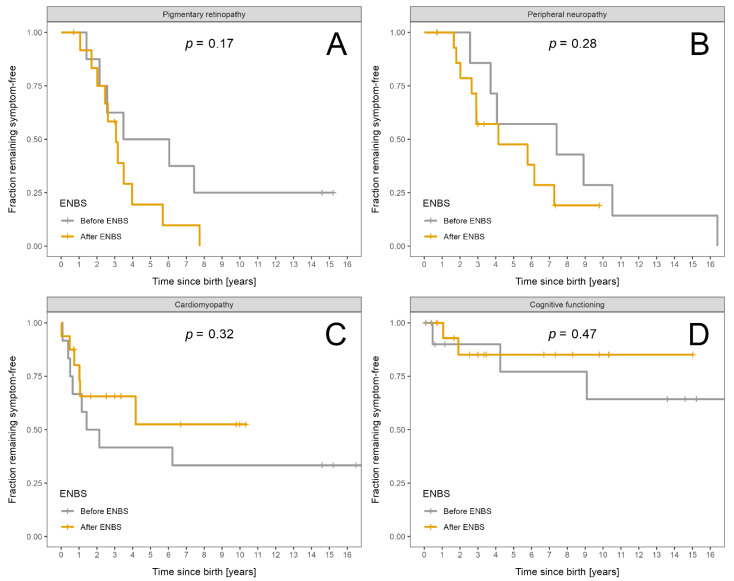
Fraction of LCHADD/MTPD patients remaining symptom free: Patients diagnosed before ENBS (*n* = 12) are shown as grey and patients diagnosed after ENBS (*n* = 16) as orange lines, respectively. (**A**) Absence of pigmentary retinopathy. (**B**) Absence of peripheral neuropathy. (**C**) Absence of cardiomyopathy. (**D**) Absence of intellectual disability. Kaplan–Meier estimator was used; *p* designates significance of difference between the cohorts.

**Figure 5 nutrients-13-02925-f005:**
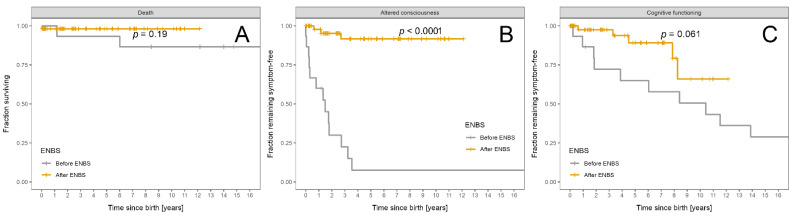
Fraction of MCADD patients remaining alive and symptom free: Patients diagnosed before ENBS (*n* = 15) are shown as grey and patients diagnosed after ENBS (*n* = 54) as orange curves, respectively. (**A**) Survival probability. (**B**) Absence of any episode of altered consciousness. (**C**) Absence of cognitive impairment. Kaplan–Meier estimator was used; *p* designates significance of difference between the cohorts.

**Table 1 nutrients-13-02925-t001:** Basic characteristics of LCHADD/MTPD and MCADD cohorts.

LCHADD/MTPD				MCADD		
	Total	Pre-ENBS	ENBS	Total	Pre-ENBS	ENBS
	*N* (%)			*N* (%)		
All patients	28 (100)	12 (100)	16 (100)	69 (100)	15 (100)	54
Male	15 (53.6)	9 (75)	6 (37.5)	34 (49.3)	9 (60)	25 (46.3)
Female	13 (46.4)	3 (25)	10 (62.5)	35 (50.7)	6 (40)	29 (53.7)
Homozygotes *	15 (53.6)	8 (66.7)	7 (43.8)	43 (62.3)	13 (86.7)	30 (55.6)
Compound heterozygotes *	10 (35.7)	3 (25)	7 (43.8)	20 (29)	2 (13.3)	18 (33.3)
Other genotypes	3 (10.7)	1 (8.3)	2 (12.5) (both MTPD)	6 (8.7)	0 (0)	6 (11.1)
	Median (range)			Median (range)		
Age at diagnosis (months)		10 (1.2–91)	<0.33		1.8 (0.1–17)	<0.33
Present age (years)		13.4 (0.1–27.8)	7 (0.7–15)		19.5 (1.2–36.6)	4.9 (0–12.1)
Follow-up (years)		9.3 (0–27.4)	7.01 (0.7–15.02)		14.57 (0.25–25.88)	4.86 (0.23–11.65)
Follow-up (patient-years)	228.4	126.6	101.8	565.1	275.6	289.5

* patients carrying the prevalent variants c.1528G > C or c.985A > G in the *HADHA* or *ACADM* genes, respectively, in homozygosity or compound heterozygosity. LCHADD/MTPD: Long chain 3-hydroxyacyl-CoA dehydrogenase deficiency/ mitochondrial trifunctional protein deficiency; MCADD: medium chain acyl-CoA dehydrogenase deficiency; pre-ENBS: cohort diagnosed before introduction of expanded neonatal screening; ENBS: cohort diagnosed after introduction of neonatal screening; HADHA: Hydroxyacyl-CoA Dehydrogenase Trifunctional Multienzyme Complex Subunit Alpha; ACADM: Acyl-CoA Dehydrogenase Medium Chain.

**Table 2 nutrients-13-02925-t002:** Acute events, chronic complications, and their severity in patients diagnosed with LCHADD/MTPD before and after ENBS.

	Total	Pre-ENBS	ENBS	*p* Value
Acute events	*N*; log-rank test (time to event)			
Patients	28	12	16	
Death	7	5	2	0.10
Altered consciousness	10	7	3	0.006
Rhabdomyolysis	21	8	13	0.053
Acute heart failure	7	4	3	0.22
Incidence of acute events	*N* of events/patient-years; χ^2^-test			
Death	7/229	5/127	2/102	0.40
Altered consciousness	24/229	15/127	9/102	0.51
Rhabdomyolysis	292/229	157/127	135/102	0.70
Severe and critical rhabdomyolysis only	37/229	13/127	24/102	0.02
Acute heart failure	10/229	5/127	5/102	0.73
Severity of acute events	Severity score/patient-years; χ^2^-test			
Altered consciousness	46/229	29/127	17/102	0.34
Rhabdomyolysis	733/229	364/127	369/102	0.13
Severe and critical rhabdomyolysis only	148/229	52/127	96/102	<0.0001
Chronic complications	*N*			
Retinopathy (patients examined)	21	8	13	
Retinopathy	17	6	11	
Neuropathy (patients examined)	23	7	16	
Neuropathy	17	7	10	
Cardiomyopathy (patients examined)	28	12	16	
Cardiomyopathy	14	8	6	
Intellectual disability (patients examined)	28	12	16	
Intellectual impairment	5	3	2	
Severity of chronic complications	AUC/patient-year- mean (SD); two-sample t-test			
Retinopathy	0.53 (0.39)	0.46 (0.36)	0.57 (0.42)	0.53
Neuropathy	0.53 (0.54)	0.82 (0.66)	0.39 (0.42)	0.07
Cardiomyopathy	0.22 (0.28)	0.33 (0.26)	0.15 (0.27)	0.09
Intellectual disability	0.05 (0.14)	0.05 (0.11)	0.06 (0.15)	0.97

For chronic complications, number of patients tested for their respective complication, i.e., retinopathy (patients examined) = number of patients who underwent an ophthalmologic examination, as well as the number of patients affected by the complication, i.e., retinopathy = number of patients affected by retinopathy are provided. AUC: Area under curve; SD: Standard deviation.

**Table 3 nutrients-13-02925-t003:** Acute and chronic complications and their severity in patients diagnosed with MCADD before and after ENBS.

	Total	Pre-ENBS	Post-ENBS	*p* Value
Acute events	*N*; log-rank test (time to event)			
Patients	69	15	54	
Death	3	2	1	0.19
Altered consciousness	16	13	3	<0.0001
Incidence of acute events	*N* of events/patient-years; χ^2^-test			
Altered consciousness	33/565	29/276	4/289	<0.0001
Severity of acute events	Severity score/patient-years; χ^2^-test			
Altered consciousness	67/565	61/276	6/289	<0.0001
Chronic complications	*N*			
Intellectual disability	16	10	6 *	
Severity of chronic complications	AUC/patient-year- mean (SD); two-sample t-test			
Intellectual disability	0.1 (0.3)	0.4 (0.5)	0.0 (0.1)	<0.0001

* One patient out of 54 not examined. AUC: Area under curve. SD: Standard deviation.

## Data Availability

The data presented in this study are available on request from the corresponding author. The data are not publicly available due to personal nature of the data.

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
