# Peer review of "Impact of Newborn Screening and Early Dietary Management on Clinical Outcome of Patients with Long Chain 3-Hydroxyacyl-CoA Dehydrogenase Deficiency and Medium Chain Acyl-CoA Dehydrogenase Deficiency—A Retrospective Nationwide Study"

_nutrients, 2021, doi:10.3390/nu13092925_

Round 1
Reviewer 1 Report
This interesting article evaluates the impact of neonatal screening for LCHAD and MCAD deficiency in Czechia.
- On page 3 the authors say that the patients included in the study were diagnosed by blood acylcarnitines and urinary organic acids and confirmed by molecular genetic testing. Up until 2011 the molecular testing included PCR/RFLP to detect the c.1528C>G and c.985A>G mutations in the HADHA and ACADM genes, respectively. As described in Figure 1 other mutations were also investigated - please clarify.
- I have difficulties in understanding the incidences of the two disorders before and after the introduction of screening. How were the incidences of the respective disorders calculated during the first 28 and second 12 years respectively.
- Patients diagnosed pre-symptomatically by family screening in affected families were excluded from the study. How many were they and were they included in the incidence numbers?
- Line 304 and 305. Fig 3, A-D and Fig 4, A-D
- The presentation of the data in the Table 1 could be simplified to be read more easily. Column 5 which is mainly a repetition of column 1 can be removed. Column 1 line 4 and 5, where the two common mutations in the respective disorders are mentioned could have an a and b and be explained in the legend to the Table as a= c.1528G
- Table 2, 3 and the Figures 3-5. The Figures need to have legends explaining how many patients were followed (equals 1 on the y-axis) in each cohort and the total number of patients in each cohort (percentage studied is not necessary since the total number of patients in each cohort is so small). This is the reason for the recommendation "Major revision" of the manuscript. Doing this, one can evaluate the Figures in a clearer way. Probability on the y-axis? I propose that you write "Fraction remaining symptom free" on the y-axis. The legend has explained that it is the fraction of the patients studied (which in turn is not always the whole cohort). The Tables 2 and 3 can then be reduced to describe the number of events/patient-years, X2-test, severity score/patient-years and severity of chronic complications etc.These two tables could be presented as supplementary material.
- Why is the age line for the patients from the screened cohort extended beyond twelve years in some of the panels of the Figures? The data is based on the observed findings during the screening period of twelve years.
- Discussion: a. Rhabdomyolysis. There was a higher incidence of rhabdomyolysis after the introduction of screening, which the authors point out could be caused by lack of complete medical records in the pre-ENBS era. Another explanation (speculation) could be that the ENBS-detected patients under-estimated the need for dietary adjustment to prevent rhabdomyolysis. This could possibly be the case, since most of them have avoided to acquire over-weight, which could be acquired by over-consumtion of glucose and glucose polymers to prevent decompensation during excersise, infections and stress.
Author Response
REVIEWER 1
This interesting article evaluates the impact of neonatal screening for LCHAD and MCAD deficiency in Czechia.
- On page 3 the authors say that the patients included in the study were diagnosed by blood acylcarnitines and urinary organic acids and confirmed by molecular genetic testing. Up until 2011 the molecular testing included PCR/RFLP to detect the c.1528C>G and c.985A>G mutations in the HADHAand ACADMgenes, respectively. As described in Figure 1 other mutations were also investigated - please clarify.
Response
Since 2011, the ACADM and HADHA genes have been analyzed by Sanger sequencing of PCR products for all coding exons (including samples previously analyzed only by PCR / RFLP, where no prevalent mutation was detected). The HADHB gene was analyzed by massive parallel sequencing in a panel of metabolic disorders.
This sentence has been added to the text section 2.1.: Since 2011 the ACADM and HADHA genes were analysed by Sanger sequencing of PCR products of all coding exons. Samples that had been analysed before 2011 and no prevalent mutation had been detected were reanalysed by Sanger sequencing after 2011.
- I have difficulties in understanding the incidences of the two disorders before and after the introduction of screening. How were the incidences of the respective disorders calculated during the first 28 and second 12 years respectively?
Response
We have revised the data thoroughly and we calculated the incidence using previously published method (Poupetova et al, J Inherit Metab Dis. 2010 Aug;33(4):387-96. doi: 10.1007/s10545-010-9093-7) in two equally large cohorts. The post-ENBS cohort included all newborn delivered in 2010-2020 period. The pre-ENBS cohort included babies born in 1999-2009 (two patients detected by a pilot screening study – see below our answer for question 7, made an exception and were not included in the calculation on the incidence rate in the pre-ENBS period), and it was expanded by adding the 1998 birth cohort to account for median age of LCHADD/MTPD diagnosis of 10 months.
Based on this revised data we modified the abstract and we have added a sentence to the Methods section 2.2.4.: Incidence rates were calculated based on the number of newborns born in 1998-2009 (1,206,358 in total, before ENBS) and in 2010-2020 (1.225,245 in total, after ENBS) as provided by the Czech Statistical Office (www.czso.cz) and the number of patients with a confirmed diagnosis (10 and 15 with LCHADD/MTPD and 9 and 53 with MCADD) who were born during these two periods, respectively.
- Patients diagnosed pre-symptomatically by family screening in affected families were excluded from the study. How many were they and were they included in the incidence numbers?
Response
They were 7 MCADD patients diagnosed by family screening. Three of them were born in between 1998 and 2009 and these were included in the calculation of the incidence rate in the pre-ENBS period. The remaining four were born before 1998.
This sentence was added to section 2.1.: Seven patients with MCADD, who were diagnosed pre-symptomatically beyond the neonatal period due to family screening in affected families in the pre-screening era, were excluded from the clinical outcome study but were included in the calculation of incidence.
4.Line 304 and 305. Fig 3, A-D and Fig 4, A-D
Response
The mistake was corrected.
- The presentation of the data in the Table 1 could be simplified to be read more easily. Column 5 which is mainly a repetition of column 1 can be removed. Column 1 line 4 and 5, where the two common mutations in the respective disorders are mentioned could have an a and b and be explained in the legend to the Table as a= c.1528G
Response
The table was simplified and modified according to the reviewer´s suggestions. The results of the biochemical analyses were removed from the table. We agree with the reviewer that the results of biochemical analyses are not essential for the publication and therefore were not mentioned in the text.
- Table 2, 3 and the Figures 3-5. The Figures need to have legends explaining how many patients were followed (equals 1 on the y-axis) in each cohort and the total number of patients in each cohort (percentage studied is not necessary since the total number of patients in each cohort is so small). This is the reason for the recommendation "Major revision" of the manuscript. Doing this, one can evaluate the Figures in a clearer way. Probability on the y-axis? I propose that you write "Fraction remaining symptom free" on the y-axis. The legend has explained that it is the fraction of the patients studied (which in turn is not always the whole cohort). The Tables 2 and 3 can then be reduced to describe the number of events/patient-years, X2-test, severity score/patient-years and severity of chronic complications etc. These two tables could be presented as supplementary material.
Response
We would like to thank the Reviewer for suggesting simplification of tables and expansion of legends to the figures. We have removed unnecessary data from both tables, modified figures and expanded figure legends as suggested. We hope that this revision addressed the major concerns of the Reviewer.
We have considered the suggestion to move the tables to the supplementary materials. We think that these tables showed hard data and we would prefer to keep the tables in the main text.
- Why is the age line for the patients from the screened cohort extended beyond twelve years in some of the panels of the Figures? The data is based on the observed findings during the screening period of twelve years.
Response
The points and/or lines beyond 12 years show lifetime data for patients detected pre-ENBS. Two patients (1 with MCADD and 1 with LCHADD) were detected by a pilot neonatal screening study that ran between 2001-2009 and were included in the post-ENBS cohort; this is the reason why their follow-up exceeded 12 years. We have modified section 2.2.4 as follows: In patients diagnosed in the pre-ENBS period there were no events beyond 16.5 years of age and curves were truncated at 17 years of follow-up. Two patients (1 with MCADD and 1 with LCHADD) were detected by a pilot screening study conducted between 2001 and 2009 and were included in the post-ENBS cohort. Therefore, the follow-up of the post-ENBS subgroup extends beyond 12 years.
- Discussion: a. Rhabdomyolysis. There was a higher incidence of rhabdomyolysis after the introduction of screening, which the authors point out could be caused by lack of complete medical records in the pre-ENBS era. Another explanation (speculation) could be that the ENBS-detected patients under-estimated the need for dietary adjustment to prevent rhabdomyolysis. This could possibly be the case, since most of them have avoided to acquire over-weight, which could be acquired by over-consumption of glucose and glucose polymers to prevent decompensation during exercise, infections and stress.
Response
Cooperation with most of our patients/parents is excellent and their compliance with the diet is good. The dietary recommendation is the same in both the pre-ENBS and the ENBS group – see 2.1 Patients section of our manuscript. Sometimes they do not respect the recommended restriction of physical activity, but there is no difference between the individual groups in this aspect.
This sentence was added to section 2.1.: Cooperation with most of our patients/parents was excellent and their compliance with the diet was good. The dietary recommendation is the same in both pre-ENBS and post-ENBS groups.
Reviewer 2 Report
Based on a before/after design, the authors intended to explore the impact of the introduction of newborn screening and subsequent clinical management on the outcome of patients with LCADD and MCADD in a retrospective approach, nationwide (Czech Republic).
Im not sure why both diagnoses were combined. I obviously understand that the diagnostic tool is the same, but the disease and its course are quite different. Although the topic is of relevance, the authors rightfully mention that there is already quite some (and stronger RCT type) of evidence on the benefits, as well as the side effects (higher incidence) of MCADD screening. Based on the current design, it is not clear how pre-screening data were collected as eg SIDS might have resulted in underrecognition, while overdiagnosis (MCAD biochemical, but will never become symptomatic). The difference in ‘phenotype’ is relevant, as the time to event was used as one of the outcome variables (cf table 1: compound heterozygotes pre and since ENBS..
I have some concerns on the fact that cases diagnosed after the neonatal period were not included in the analysis, as this – in the post implementation era – are missed cases, or have I misunderstood this sentence (line 142-144, section 2.2.)
Table 1: C16 and C18 concentrations, I may have missed this, but collected ? when
Figure A2-B: visual inspection was used to assed the individual weight to the weight trends, but the figures could also be analysed using the individual Z scores, as the picture suggest that the case had a Z score > 0 (figure 2B>2A). Can the authors elaborate on this, as this might have long term negative effects, of relevance as the diagnosis has resulted in a higher incidence ?
Figure numbers in the last sentence of pg 9 are likely wrong (should read 3 A-D and 4 A-D ?)
For the MCADD, I miss data on the ‘burden of the diagnosis’ like the number of admissions ?
I agree that the small cohort has resulted in (too) extensive analysis and interpretation (LCADD), so that pooled analysis of national cohorts is the only way to fully assessment the benefits ?
too what extent has the final analysis been impacted by the absence of availability to triheptanoin (but one case), and that screening implementation without access to the perceived more effective interventions is an issue (cf https://www.researchsquare.com/article/rs-41739/v1). Along these lines, are there any ‘first world’ countries where screening in not yet done ?
Author Response
REVIEWER 2
Based on a before/after design, the authors intended to explore the impact of the introduction of newborn screening and subsequent clinical management on the outcome of patients with LCADD and MCADD in a retrospective approach, nationwide (Czech Republic).
Im not sure why both diagnoses were combined. I obviously understand that the diagnostic tool is the same, but the disease and its course are quite different. Although the topic is of relevance, the authors rightfully mention that there is already quite some (and stronger RCT type) of evidence on the benefits, as well as the side effects (higher incidence) of MCADD screening. Based on the current design, it is not clear how pre-screening data were collected as eg SIDS might have resulted in underrecognition, while overdiagnosis (MCAD biochemical, but will never become symptomatic). The difference in ‘phenotype’ is relevant, as the time to event was used as one of the outcome variables (cf table 1: compound heterozygotes pre and since ENBS..
- I have some concerns on the fact that cases diagnosed after the neonatal period were not included in the analysis, as this – in the post implementation era – are missed cases, or have I misunderstood this sentence (line 142-144, section 2.2.)
Response
In our study we aimed at comparing the outcome of patients diagnosed by ENBS early in life with those diagnosed later due to clinical symptoms. The group of patients ascertained by family screening did not fit any of these two categories and we have excluded them from evaluation of the efficacy of ENBS.
Section 2.2. was adjusted as follows: For comparison of clinical outcome LCHADD/MTPD and MCADD patients were subdivided into those clinically ascertained in the pre-ENBS period and those diagnosed pre-symptomatically in the neonatal period by ENBS.
Section 2.1. line 104-106: Seven patients with MCADD, who were diagnosed pre-symptomatically beyond the neonatal period due to family screening in affected families in the pre-screening era, were excluded from the clinical outcome study…
- Table 1: C16 and C18 concentrations, I may have missed this, but collected? when
Response
Acylcarnitine levels were determined from dried blood spots at the time of diagnosis. However, Reviewer 1 suggested that the results of biochemical analyses are not essential for the publication and we have removed these data from the table.
- Figure A2-B: visual inspection was used to assed the individual weight to the weight trends, but the figures could also be analysed using the individual Z scores, as the picture suggest that the case had a Z score > 0 (figure 2B>2A). Can the authors elaborate on this, as this might have long term negative effects, of relevance as the diagnosis has resulted in a higher incidence?
Response
We thank the Reviewer for spotting this omission. Stimulated by the above criticism we re-checked Figure 2 and discovered that data from several patients were loosed during statistical analysis. We apologize for the mistake and provide corrected figures. In addition, we decided to use BMI together with the Z scores instead of weight in order to illustrate the burden of overweight more clearly. Results and discussion were adjusted accordingly.
Section 3.3.: Height and BMI during follow-up of patients with LCHADD/MTPD were plotted and compared to normal values. Height of 8 patients from both subgroups was below -2 Z scores, which was related to prematurity in six of them as they exhibited a sufficient catch-up growth within the first year of life. Only two children remained below -2 Z with respect to height during long-term follow-up. BMI exceeded +2 Z in 5 children (with maximum of +5.6 in one girl) during long-term follow-up.
Figure 2 legend: Anthropometric parameters in patients with LCHADD/MTPD. (A). Height of patients diagnosed before ENBS (data available in 10 patients, 111 measurements, median 11 measurements per patient) and by ENBS (n = 16, 95 measurements, median 6.5 measurements per patient) during follow-up compared to reference range. (B). Body Mass Index of patients diagnosed before ENBS (n = 12, 111 measurements, median 11 measurements per patient) and by ENBS (n = 16, 95 measurements, median 6.5 measurements per patient) during follow-up. Patient data are shown as individual points, median and 2-98th centile of sex-adjusted reference ranges for the Czech population are shown as solid and dotted lines, respectively.
Discussion: We observed good growth with only two patients below normal range and weight control in these patients. Five (5/28) patients had a Z score for BMI > +2, which is lower than in a Swedish study that reported overweight or obesity in the majority (8/10) of patients.
- Figure numbers in the last sentence of pg 9 are likely wrong (should read 3 A-D and 4 A-D?)
Response
Thank you, the mistake was corrected.
- For the MCADD, I miss data on the ‘burden of the diagnosis’ like the number of admissions?
Response
We focused only on emergency admissions due to hypoglycaemia associated with impaired consciousness because we were confident about completeness of this type of data for patients from both pre- as well as post-ENBS era. The results are shown in Table 3 (N of events/patient-years – Altered consciousness).
We agree with the reviewer that analysis of the burden of all admissions would have been interesting. Unfortunately, we are unable to collect data on all admissions in local hospitals for other problems including short-term admission to prevent decompensation during acute infections. Due to this paucity of data, we cannot provide such data with reasonable confidence.
- I agree that the small cohort has resulted in (too) extensive analysis and interpretation (LCADD), so that pooled analysis of national cohorts is the only way to fully assessment the benefits?
Response
The aim of our study was to assess the nationwide impact of ENBS on clinical outcome. However, we still think that such national data, even in this limited scale, are interesting and have a value of their own, in particular in terms of future research within the framework of international collaboration with the possibility of conducting metanalyses of larger cohorts of patients. Some of the statistical methods used in our manuscript for evaluation of chronic complications may be useful for such studies in the future.
- too what extent has the final analysis been impacted by the absence of availability to triheptanoin (but one case), and that screening implementation without access to the perceived more effective interventions is an issue (cf https://www.researchsquare.com/article/rs-41739/v1). Along these lines, are there any ‘first world’ countries where screening in not yet done?
Response
While triheptanoin helps with the muscular complications, cardiomyopathy, and hypoglycemia associated with LCHAD deficiency (Guggon N et al., Mol Genet Metab. 2021), it does not appear to impact the eye disease and development of peripheral neuropathy. Screening of MCADD is not available e.g. in Belorus, Bulgaria, Cyprus, France, Greece, Litvia, Russia, Serbia and Turkey; of LCHADD also in Belgium, Luxembourg, Switzerland and UK. For more detail please see Loeber JG et al., Int.J.Neonatal Screen. 2021.
The following paragraph was added to the discussion: Due to lack of evidence for unambiguous benefit of ENBS for LCHADD/MTPD patients some countries (e.g. Belgium, Luxembourg, Switzerland and the UK) have not included LCHADD/MPTD in their neonatal screening programs [Loeber JG et al., Int.J.Neonatal Screen. 2021]. Future routine use of the more efficient MCT oil, triheptanoin, may result in a better outcome of LCHADD/MTPD patients than reported in our study [Guggon N et al., Mol Genet Metab. 2021].
Citation Loeber JG et al., Int.J.Neonatal Screen. 2021 and Guggon N et al., Mol Genet Metab. 2021 have been added to references.
Round 2
Reviewer 1 Report
The manuscript has been improved. I still think that Figure 4A and B are a bit misleading, since the number of patients investigated is not 12 and 16 for the pre- and post ENBS cohorts, respectively, for these parameters. This is, however, presented in the corresponding Table.
Reviewer 2 Report
no additional comments,